# Effectiveness of barber-facilitated "Doing What Matters in Times of Stress" intervention among urban literate youths in Western Kenya: A cluster randomised trial

Protus Musotsi Yabunga[1,2,3]*, Phiona Naserian Koyiet[4], Harriet Musimbi[1], Ken Simiyu[1], Elijah Chemorei[1], Peter Koskei[5], Stella Evangelidou[2], Mary Nangukhula[1]

1 Sentum Scientific Solutions, Nairobi, Kenya, 2 Barcelona Institute for Global Health (ISGlobal), Barcelona, Spain, 3 Facultat de Medicina i Ciències de la Salut, Universitat de Barcelona (UB), Barcelona, Spain, 4 World Vision International, Nairobi, Kenya, 5 Moi University, School of Public Health, Eldoret, Kenya

* protusmusotsi@gmail.com

## Abstract

Youth with mental health disorders are a growing concern in Kenya, yet services are inaccessible. "Doing What Matters in Times of Stress (DWM)" is a scalable self-help intervention that can be delivered by lay providers. However, its effectiveness in low-resource, non-humanitarian settings is unknown. We evaluated the effectiveness of barber-facilitated DWM intervention in reducing psychological distress, while improving functioning and resilience among literate urban youths in Western Kenya. This cluster randomised trial involved 15 barbers in intervention group and 15 in the waiting-list control. Barbers were from 30 barbershops in Bungoma and Kitale towns, aged 18–30 years and literate in English. Using mental health-themed music as an entry point, barbers recruited youth aged 18–29 years, fluent in English, and had mild or moderate psychological distress and functioning difficulties. Intervention participants received DWM guide and/or audios, supported by barbers through three individual sessions. Waiting-list control participants received the intervention after five weeks. Outcome measures included Patient Health Questionnaire (PHQ-9), Generalised Anxiety Disorder-7 (GAD-7), Perceived Stress Scale (PSS-10) and psychological outcome profiles (PSYCHLOPS). We analysed data using cumulative ordinal multilevel models. Between 31st July 2023–27th January 2024, 330 eligible youths were enrolled (n = 158, from 11 intervention arm barbers and n = 172 from 10 control barbers). Post-intervention, intervention group youth had 95%, 82% and 71% lower odds of higher depression (AOR = 0.05, 95% CI 0.02-0.10, p < 0.001), anxiety (AOR = 0.18, 95% CI 0.09–0.34, p < 0.001) and stress (AOR = 0.29, 95% CI 0.12–0.71, p = 0.007), respectively. They also had 86% lower odds of higher self-identified problem severity scores (AOR = 0.14, 95% CI = 0.08 -0.27, p < 0.001) and 4.3 times higher odds of higher resilience levels (AOR = 4.33, 95% CI 2.00–9.36, p < 0.001).

**Data availability statement:** All relevant data are within the paper and its Supporting Information files.

**Funding:** This project was funded by the Grand Challenges Canada Global Mental Health Program Grant [R-GMH-POC-2211-59472] with the financial support of the Government of Canada provided through Global Affairs Canada, awarded to Sentum Scientific Solutions (PMY, HM, KS, EC, and MN are employees). The funders had no role in study design, data collection and analysis, decision to publish, or preparation of the manuscript.

**Competing interests:** The authors have declared that no competing interests exist.

DWM effectively reduced psychological distress and improved resilience among youths. Barbershops are promising community places for mental health promotion. Long-term, multi-site studies are required to assess outcome sustainability and scalability.

**Trial registration**: Pan African Clinical Trials Registry PACTR202306502042812

## Background

Young people globally are believed to be disproportionately affected by mental health disorders, and this is often associated with important life changes, which encompass productive years for education, employment and social participation [1]. Approximately one in every six young adults (15.5%) in Europe and one in four in Sub-Saharan Africa (26.9%) are estimated to suffer from a mental health disorder [2].

Kenya is among the top ten African countries with the highest prevalence of depression (4.4%) [3]. Approximately one in every four Kenyans seeking healthcare from a health facility suffers from a mental disorder [4]. Depression, substance abuse, and anxiety are the most frequent mental health conditions diagnosed in Kenya [5]. Despite the scarcity of disaggregated data for youth mental health in Kenya, a high prevalence of mental health disorders among adolescents of up to 46% and 38% for clinically significant depression and anxiety, respectively, has been previously reported [6].

While mental health disorders are an increasing area of concern in the country, limited mental health services exist due to shortages of professional mental health service providers. Besides, young people rarely utilise facility-based mental health services in the country. The few who do so seek services when it is already late, with a mean time of 16.6 months from the onset of symptoms to accessing care at a clinic [7]. Access barriers to services include lack of awareness regarding mental health and mental illness, community perceptions of mental illness as synonymous with being bewitched, stigmatisation of mental conditions [8], limited funding, scarcity of mental health service providers and lack of local mental health plans [9]. These challenges necessitate culturally acceptable, cost-efficient, community-based mental health care services. Such interventions should first address raising awareness on mental health and targeting stigma, among other access challenges, with the view to promote mental health and prevent severe mental health conditions.

Community-based mental health interventions are a pivotal aspect of global mental health strategies, particularly in addressing the mental health needs in low and middle-income countries (LMICs) and marginalized communities [10]. Understanding the effectiveness of such interventions is therefore critical to ensure a sustainable impact on individuals and their communities. "Doing what matters in times of stress" (DWM), a core part of the Self-help Plus (SH+) [11], is a psychological intervention developed to address some of the highlighted challenges and with the ability to be rolled out rapidly. It is accessible, easy to use, and easy to scale-up. It can be

delivered as a self-directed intervention or supported by lay providers. This makes it suitable for our Western Kenya settings with increasing mental disorders among youth, but limited mental health professionals and facilities, and insufficient public investments in mental health countrywide [9].

DWM involves an illustrated self-help stress management guidebook and brief pre-recorded audio exercises with five sections covering psychoeducation on stress and the core strategies from the SH+ intervention (grounding, unhooking, acting on your values, being kind and making room) [12]. It is based on acceptance and commitment therapy (ACT). It was designed to support learning during SH+, but it can be used as a standalone intervention with or without support from a trained helper [11]. Section one of the guide focuses on grounding, bringing attention back to the present moment when caught up in distressing emotions. Section two is on unhooking, which involves noticing difficult thoughts and feelings, naming them, and refocusing. Section three is on acting on own values, which consists of identifying personal values and taking actions to live within these values. Section four is on being kind, which involves enhancing and encouraging kindness towards oneself and others. Section five focuses on making room and learning to tolerate stress while acting consistently with values [11].

Beyond DWM application as a core intervention in humanitarian settings [13] and the existing literature supporting the effectiveness of the broader SH+ package [14–17], limited data exists about DWM effectiveness per se [18] and its applicability in non-humanitarian, low-resource settings.

The study aimed to determine the effectiveness of DWM intervention guided by lay facilitators in reducing psychological distress and improving functioning and resilience among young people in Kenya. All outcomes were individual participant level outcomes. We used a cluster randomised design for practical reasons to avoid cluster level contamination.

## Materials and methods

### Ethics statement

The study protocol was reviewed and approved by the Moi University/Moi Teaching and Referral Hospital Institutional Ethics Review Committee (IREC) (Approval number: 0004378). As required by the Kenyan Science, Technology and Innovations Act of 2013, a research permit for the study was granted by the National Commission for Science, Technology, and Innovations (NACOSTI) (Licence number: NACOSTI/P/23/25082). Written informed consent was obtained from each participant before recruitment into the study. The data was pseudo-anonymised, with participants' and barbers' identifiers replaced with unique numbers.

### Study design

We used a cluster randomised design, allocating barbers, as lay helpers, in the study arms (intervention-waiting list control) instead of individual participants to avoid cluster level contamination, considering youth from the same place or network were likely to visit the same barbershop. The study was registered prospectively at Pan African Clinical Trials Registry (PACTR202306502042812).

### Study settings

We selected Bungoma and Trans-Nzoia counties for the study due to the shortage of mental health providers and services in both counties. Only four public health facilities in Bungoma and one in Trans-Nzoia included established mental health services since 2020 [9]. Trans-Nzoia county had two psychiatrists, five psychiatric nurses, and twelve psychologists/counsellors [9] despite having a total population of 990,341 of which 277,601 are youths aged 15–29 years [19]. There were only one psychiatrist, six psychiatry nurses, six psychiatric clinical officers and two psychologists/counsellors in Bungoma County [9] which has 1,670,570 people of which 464,188 are young people aged 15–29 years [19]. None of the two counties had a provision for a mental health-specific budget [9]. We intentionallyfocused the project in the major towns of the two counties (Bungoma and Kitale towns), where most youths are concentrated.

We selected barbershops as community places since they are usually frequented by young people and play an important role in young men's and women's grooming, as part of the local culture. Fifty barbershops were visited by the project coordinator and/or one of the ten research assistants. Information about the project was shared with barbers, who were then asked to share the project information with fellow barbers through their WhatsApp groups and networks. Those interested had to contact the project coordinator. The barbershops were selected based on the owners' willingness to participate in the study. A total of 42 barbers in both counties showed interest, 30 out of 42 attended all the training sessions and were recruited into the study.

## Participants

Of the 42 barbers initially selected, we included 30 barbers in 30 different barbershops from Bungoma (n = 15) and Kitale (n = 15) towns in Western Kenya. Inclusion criteria were: oral and written fluency in English, being aged 18–30 years, informed consent and approval to participate in the study by the respective barbershop owners (in cases where the recruited barbers were not the owners), barbers' availability to attend a 5-day training and acceptance to work voluntarily for the project, dedicating at least one hour a day to the intervention activities during work hours or after their active work shift during a period of six months. The barbershop had to be staffed with more than one barber, so that the project could not interfere with the normal running of the barbershop, when one of them was involved in the project intervention.

We included youth aged 18–29 years visiting the selected barbershops for routine haircuts/ hairdressing or to relax, as is the cultural norm in Western Kenya. The eligible young participants had fluency in English language speaking and comprehension, mild or moderate psychological distress based on the screening results (PHQ-9 score ≥ 5 to <15) and experienced functioning difficulties in various areas of life, as indicated using WHODAS with a score ≥ 17. The PHQ-9 cut-off points of 5–14 were used since this interval identifies individuals with mild and moderate psychological distress [20]. The WHODAS cut-off point of 17 was applied since it identifies the 90th percentile of functioning impairment in populations across countries [21]. Besides, this cut-off has previously been used in related interventions in Kenyan settings [22].

We excluded those with imminent suicide attempts or suicide ideation, those with a neurological or severe mental disorder, substance use disorder or cognitive impairment as screened using a short self-reported questionnaire and an observation assessment grid [23] completed by the research assistants. We provide the questionnaire in Supplementary S1 Text. We also excluded those who did not consent to participate in the study and those who planned to relocate from the area within two months after their first contact with the barber.

## Sample size

Initially, we planned to enrol 300 participants in intervention and 300 in control arms involving 20 clusters (i.e., barbers) per arm. 15 participants were needed for each cluster to detect a small effect size, Cohen's d = 0.2 considering intra-cluster correlation coefficient of 0.01, a type I error rate of 5%, and a power level of 80%. With some of the initially recruited 42 barbers dropping out or not being eligible, our biostatistician recalculated the sample size using the National Institute of Health Research Methods Resources for cluster-randomised trials [24]. We considered an intra-cluster correlation coefficient of 0.01, a type I error rate of 5%, and a power level of 80%. Eleven groups (clusters) per study arm, with 15 people per cluster, were needed to detect a medium effect size of Cohen's d = 0.49. (S1 Table).

## Allocation

We used a simple random allocation approach to allocate barbers to the intervention or waiting- list control arms. During the last day of the barber training in each town, project collaborators independent to the study, wrote numbers on pieces of paper ranging from 1 to 15 for the first town and 16–30 for the second town. The pieces of paper with the numbers were folded, put in a bowl, and mixed. The barbers were asked to pick one each. Barbers who selected even numbers were

placed at the intervention arm, while those who selected odd numbers were assigned to the control group. Eligible participants at control group were offered the intervention after a five-week waiting period.

## Allocation concealment

Due to the nature of the intervention, it was impossible to blind the barbers and the participants. Research assistants were not informed of the barbers group allocation.

## Study procedure

Recruitment of participants to our project, referred to as "We Matter Too", was done from 31st July 2023 to 27th January 2024. The whole trial took place from 31st July 2023 up to 15th April 2024 when the follow-up of last study participant was completed. The project was a multi-stage implementation project that included: (a) identification of barbershops, (b) recruitment and training of barbers, (c) composition of youth-friendly songs on mental health by local musicians and their delivery at the recruited barbershops, (d) recruitment of young people/clients of barbershops, and random allocation of barbers to intervention and control arm groups, (e) intervention group: delivery of intervention material to young participants and organisation of three face-to-face sessions. We provide a detailed description of the study procedure in the supplementary S2 Text.

**Intervention delivery.** The trained barbers approached the youth while visiting the barbershop for their routine haircut/hair plaiting or chit-chatting, a common practice in the region. The barbers opened a conversation on mental health using as a trigger, the songs composed for the purpose of the study, before introducing information about the intervention. They referred those interested in the intervention to the research assistants for consenting and screening. Those with scores indicative of severe depression symptoms or other mental health conditions were referred to the nearest health facility or institution offering mental health care. The majority of the research assistants had a background in psychology and made the referrals as needed. For those with no psychology background, an internal referral to the project clinical psychologist was made for further external referral to specialist care.

The eligible participants were sent back to the barbers who recruited them. In the intervention group, the trained barbers met the participants in three face-to-face sessions, for a maximum of one hour each at three intervals of the intervention.

First session: After informed consent and pre-intervention assessment by research assistant, the first session took place at the barbershop, or outdoors provided there was privacy, and lasted approximately one hour. It covered an introduction to the DWM stress management illustrative guide and audio, an open discussion based on young person's needs, delivery of the DWM intervention materials, planning on its use, and planning of subsequent meetings.

One-week follow-up: During this follow-up meeting, the barber met the participant at the young person's place of choice, provided there was privacy and the barber was also comfortable with this choice. They discussed progress in using the DWM stress management guide and addressed any challenges faced. The barber helped the participant understand any unclear concepts by practical examples, and addressed any difficulties in following the self-help techniques. The session lasted approximately 40 minutes.

Five-week follow-up: After one month (5th week of intervention), the barber had another face-to-face follow-up session with the participant to assess his/her experience overall using the intervention materials, and further, help them understand any remaining unclear concepts, while addressing any practical difficulties. This final meeting included a discussion and planning on how the participant could continue using the intervention booklet/audio without the barber's support. This last session lasted approximately one hour. The research assistant did a post-intervention assessment after this third session.

**Waiting list arm procedure.** After recruitment and assessment, eligible participants in the waiting-list control arm were informed by their barber to come back after five weeks for the start of the intervention. After the five-week waiting period,

participants in the waiting list control were sent to the research assistant for follow-up assessment, after which they received the intervention materials from the barber. The barber also followed-up on them after one week and five weeks as was the case with the intervention arm participants.

## Outcome assessment and follow-up

The pre- and post-intervention assessments for both intervention and control groups were performed by research assistants on KoboToolbox [25]. Before pre-assessment, the research assistant explained the objectives and methodology of the project to the eligible youths and received written informed consent from them before administering the questionnaires.

The project coordinator and project clinical psychologist followed-up with the barbers weekly to check their fidelity to the intervention and address their difficulties, ensuring appropriate intervention delivery.

## Outcome measures

The primary outcome was psychological distress. For the purpose of this study, we defined psychological distress to consist of depression, anxiety, self-perceived stress and the emotional impact of personally-identified problems. We measured (i) depression through Patient Health Questionnaire (PHQ-9) [20,26], (ii) anxiety using Generalised Anxiety Disorder-7 (GAD-7) Scale [27], (iii) stress through Perceived Stress Scale (PSS-10) [28], and (iv) personally identified problems using the Psychological Outcome Profile (PSYCHLOPS) [29,30]. Secondary outcomes included functioning impairment measured through the administration of the 12-item WHO Disability Assessment Schedule 2.0 (WHODAS 2.0) [21] and resilience measured through the Brief resilience scale (BRS) [31]. All tools had been tested for reliability and validity or used in the Kenyan context. We provide more details on the tools, their scoring and validation (S3 Text).

## Data analysis

We analysed the data using R [32] and R Studio [33]. We used intention-to-treat approach, where we analysed all participants in their assigned group, regardless of their level of engagement with the intervention.

For descriptive analysis, we conducted crosstabulation to compare the categorical sociodemographic variables in the two study arms. We used the chi-square test to determine the significance of differences in the categorical demographic variables (gender, town of residence, level of education, employment status and marital status) and categorised pre-test ordinal outcome variables (depression, anxiety, stress, and resilience) between the two study arm groups. We used Wilcoxon Rank Sum Test to test for significance of differences in the distributions of age, WHODAS scores and PSYCHLOPS scores between the intervention and control group participants.

We determined significant changes in outcome measures per participant within each study arm. We classified PHQ-9 total scores into four categories with a score of 0–4 indicating no depression, 5–9 representing mild depression, 10–14 representing moderate depression and any score of 15 and above representing severe depression [20]. For GAD-7 scores, we categorised them into 0–4 (no anxiety), 5–9 (mild anxiety), 10–14 (moderate anxiety), and 15–21 (severe anxiety) [27]. We categorised PSS-10 scores as 0–13 low stress and 14–26 moderate stress and 27–40 high stress [28,34]. For BRS, we considered a score of 1.00–2.99 as low resilience, 3.00–4.30 normal resilience and 4.31–5.00 high resilience [35,36].

Any participant, whose outcome scores changed from a higher severity category to a lower one at post-intervention, was considered to have improved, as agreed by the project psychologists and Mental health and Psychosocial Support (MHPSS) experts. A change from a lower severity category at pre-intervention to a higher severity category post-intervention showed that the participants' condition had worsened. Additionally, anyone with the same score categories at post-intervention and pre-intervention were regarded as experiencing no significant change.

With lack of validated categorization of WHODAS and PSYCHLOPS, we used pragmatic approach to scale their total scores into ordinal outcome variables. We divided the individual WHODAS and PSYCHLOPS total scores by 12 and 4 respectively, the number of scored items that make up each of the two tools.

We first fitted multilevel linear mixed effect model to our outcome scores. However, our outcomes and resulting model outputs did not meet the model assumptions of linear relationship between predictors and outcome and normality of residuals. Besides, our outcome scores were not continuous variables in nature. Hence, we did not consider this model appropriate.

We used ordinal models instead, since ordinal outcome assessment tools were administered. The resulting scores from such tools have been shown to routinely violate the normality distributional assumption of linear models [37]. The use of linear models treats ordinal data as equal interval values, when in practice, differences from one ordinal category to another vary [37,38]. For example, for PHQ-9 tool, responses can vary as Not at all = 0, Several days = 1, More than half the days = 2 and Nearly every day = 3, in an ordinal way, though the psychological distances between them are not guaranteed to be equal. Treating scores from such scales as if normally-distributed equal-interval continuous values is considered inappropriate [37]. Such practices can result in systemic errors, inflate type I error and distort estimate of effect size [37]. Use of linear or nominal models for such data leads to distortion of the result, hence it is recommended to use ordinal models for such datasets instead [37,38]. This informed our choice for ordinal models for our data analysis.

For univariate and multivariable analysis, we first fitted a cumulative link mixed model (CLMM) using the clmm function from the ordinal package in R. This is an ordinal mixed-effects regression model that accounts for the clustering in the data, by incorporating random effects [39]. We used 10 quadrature points for a more accurate approximation of the likelihood when estimating the random effects.

We fitted the model to our outcomes, one at a time, controlling for town of residence, gender, level of education, and employment status as covariates while using barbers (clusters) as a random effect to account for the clustered nature of our dataset. Our effect size of interest was the interaction between time (post intervention) and allocated group (Intervention group), which tested whether changes in our outcome over time differed between the intervention and control group. We reported the odds ratio (OR) obtained by exponentiating the model estimates and its associated 95% Confidence Interval (CIs). We considered a p value <0.05 as statistically significant.

To assess ordinal model assumptions of proportional odds, we used Bayesian regression mixed models using 'Stan' based on brm function from the brms package in R [40]. This model allows for fitting of cumulative, unequal variance and category specific effect models, while accommodating the multilevel structure of the data [38], hence allowing us to test for assumptions and identify the best fit model for our dataset, whose features are not supported in ordinal package. To control the Markov Chain Monte Carlo (MCMC) sampling process used to approximate the posterior distributions of the model parameters, we used 4 independent MCMC chains, 4 Central Processing Unit (CPU) cores for parallel processing, 4000 iterations per chain to increase the reliability and accuracy of the posterior estimates, and 2000 warmup iterations in each brm model.

We first fitted the Bayesian cumulative ordinal regression model for each of our outcomes (Models1) accounting for study covariates and cluster random effect. We provide the models result for each outcome in S1 File. We used marginal effect plots of predicted probabilities to visualise the relationship between each predictor and each of the outcome ordinal categories checking for consistency across all levels as illustrated in the tutorial by Bürkner and Vuorre [38]. Significant variation in predictor effects across outcome categories was used as indicative of proportional odds assumption violation.

We fitted category specific effect models, Models2 (Models in S2 File) which relaxes the proportional odds assumption, hence allowing each outcome levels to be affected differently by the covariate based on individual mental health outcome rating category. As cumulative models do not support this for models with random effect, we used adjacent-category model using family = acat() instead of family = cumulative() in the brm model. We wrapped the study independent variables in cs() [38].

We fitted other cumulative models 3,3a and 3b for each outcome (Models in S3, S4 and S5 File) with a discriminatory parameter (disc) while allowing for unequal variance across outcome categories, assuming unequal variance only for variables noted to have unequal variance (Models in S3 File), assuming unequal variance in all covariates (Models in S4 File) and assuming unequal variance in all covariates including between interaction of time and allocated group (Models in S5 File), hence partly relaxing the proportional odds assumption [38].

As we fitted the category-specific effect model using the adjacent-category model, we fitted an additional adjacent-category model without category-specific effects (Models in S6 File) to ensure that any difference observed between models 1 and 2 was not due to comparing two models from different classes. We compared our models using their leave-one-out information criterion (LOOIC), aimed to determine the best fit model for the data as demonstrated in the tutorial by Bürkner and Vuorre [38].

In selecting best fit model, we considered those that had smallest LOOIC, whose difference from the other models was twice their corresponding standard errors [38]. None of our model met this criterion indicating that our models did not have significant differences in terms of best fit. While category-specific effects models using adjacent-category method had lower LOOIC values, than the other models, they did not meet the above criteria and were unstable with convergence issues, not being able to produce LOOIC values in some instances as was the case for depression and resilience. Proportional odds assumption models (Models in S1 File) had nearly equal LOOIC to unequal variance models for stress, functioning and resilience outcomes (S2 Table). With no significant fit differences, we reported the equal variance model as our main outcome model. As there were no significant differences between the results of the Bayesian equal variance models in S1 File and those obtained from the cumulative ordinal multilevel model using the clmm function from the ordinal R package, we report the univariate and multivariate results from the cumulative clmm model as our main results. We present the results of the Bayesian models1 in S1 File in the sensitivity analysis and the raw results of models 1, 2, 3, 3a, 3b and models 4 as part of the supplementary material (Models in S1, S2, S3, S4, S5 and S6 File).

We visualised the data for missingness patterns. We tested if the data was missing completely at random (MCAR) using the TestMCARNormality function from the MissMech package [41]. We used missing_compare() function from the finalfit package [42] to test if missingness for each of our outcome variable was associated with any of the explanatory variables in our study. Several of our explanatory variables significantly predicted missingness in our outcomes, suggesting plausibility of MAR assumption. For example, missingness in depression scores was significantly associated with allocation group, time point of assessment and town of residence.

We also used logistics regression to model the probability of missingness for each outcome as a function of the explanatory variables in the study. For each outcome, we created a binary variable missing outcome, "true" if outcome score was missing, "false" otherwise. We used this outcome for the logistics regression. A number of the explanatory variables were significant predictors of missingness. For example, for all outcomes, allocated group and town of residence were significant predictors of missingness, supporting plausibility of MAR assumption.

Hence, we considered our missing post-assessment data to be MAR. We imputed post-intervention data for participants lost to follow-up using multiple imputation technique. We considered the clustered nature of our data in the imputation using the mice packages in R [43]. We used the two-level predictive mean matching method (2l.pmm) as it accounts for the hierarchical nature of the clustered data during imputation. We imputed the total scores instead of individual outcome items. This approach is recommended when all the items of the outcome have missing data [44], as it was our case where post-assessment outcome data for people lost-to-follow-up was missing. We imputed data using 50 imputations and 20 iterations. We then fitted ordinal cumulative link mixed models on the pooled imputed data to determine the effect size.

As part of additional robustness checks for our study results, we assessed how changes in random effect and cluster sizes affected our study outcomes. We first evaluated how our results varied when (i) our model only included clusters as the only random effect, (ii) both cluster and individual participant as random effects and (iii) when no random effect was included. Secondly, we excluded study clusters with less than 5 participants, followed by exclusion of clusters with less than 10 participants and assessed the changes in our model output.

## Results

We randomised 30 barbers from 30 different barbershops into the intervention arm (n = 15) and waiting list control arm group (n = 15). During the project, 4 barbers in the intervention and 5 barbers in the waiting-list control arm did not recruit any eligible participant. A total of 158 participants were included in the intervention group recruited by 11 barbers and 172 in the waiting-list control group from 10 barbers. A total of 14 participants in the intervention group and 47 in the control group were lost-to-follow-up (Fig 1).

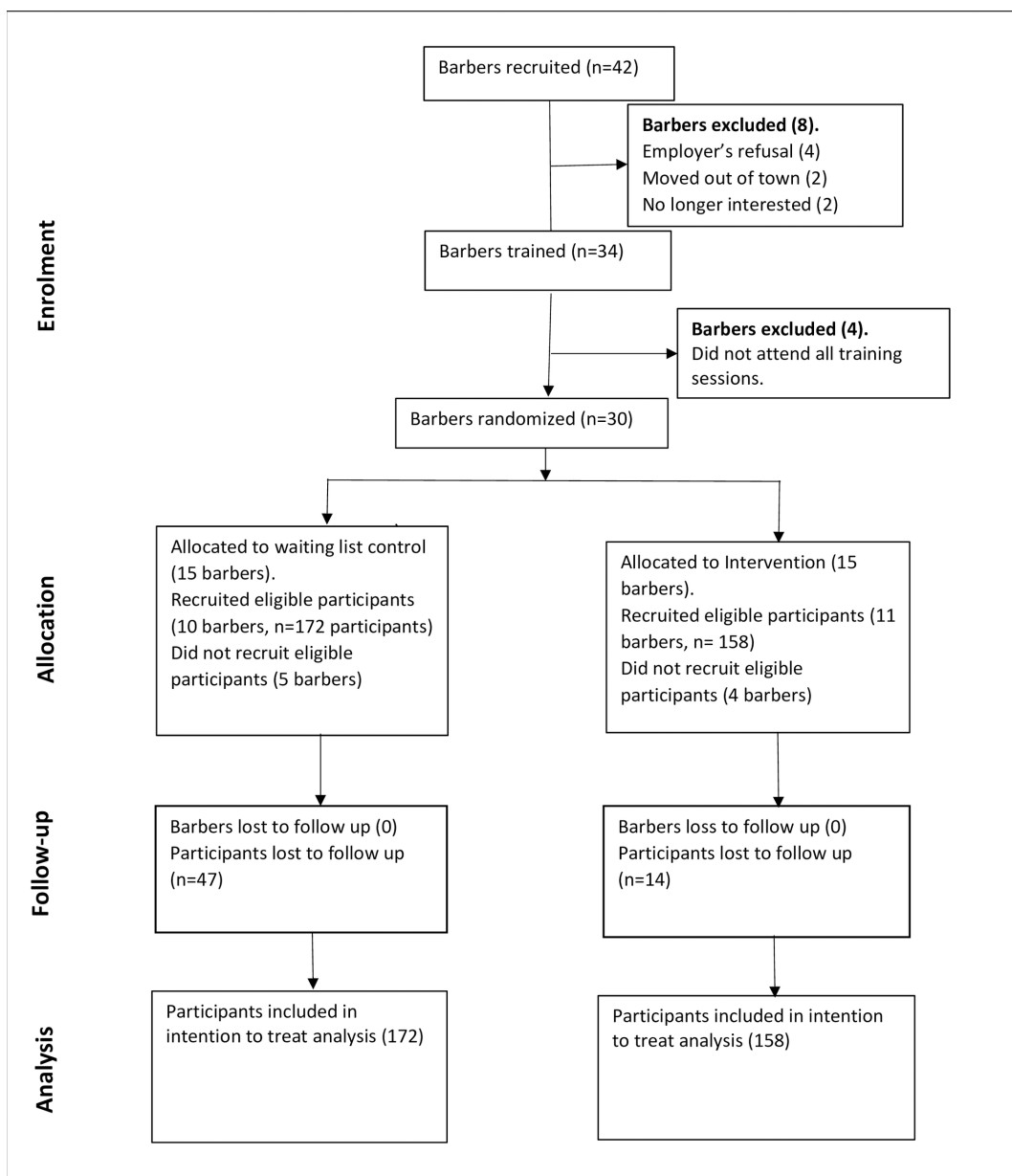

**Fig 1. Study flow diagram showing the study design, enrolment, intervention allocation, follow-up and data analysis.**

## Participants' characteristics

Among the 330 eligible young people, 168 (50.9%) were from Bungoma town, and 162 (49.1%) were from Kitale town. More than half were males (n = 176 (53.3%)) and the majority were single (n = 233, 70.6%). Half of them, 166 (50.3%) had some form of employment or were involved in an income-generating activity, mostly informal jobs. There were no significant differences in the proportion of participants in the intervention and control groups regarding town of residence, age, and marital status. However, significant differences existed between participants in the two groups regarding gender, level of education and employment status (p < 0.05). (Table 1).

## Pre-intervention outcome levels

There were no significant differences between the intervention and control arm participants at pre-intervention for depression, anxiety and resilience levels. The levels of stress were significantly different between the intervention and control group (p < 0.001). A high proportion of participants in the control group (38.4%) had higher level of stress compared to intervention group (19.0%) (Table 2).

## Unadjusted absolute effect of the intervention

There was a significant improvement of depression symptoms among 65.3% of the participants in the intervention arm compared to 21.4% in the control arm. There was also a high proportion of participants whose depression level worsened in the control group (17.5%) compared to the intervention group (6.9%). The changes in depression levels were significant (p < 0.001). The intervention resulted in significant improvement of anxiety symptomatology in 53.5% of the participants at the intervention arm compared to 28.0% at the control arm. The intervention also resulted in stress improvement in 32.6% of the participants at the intervention arm compared to 25.6% at the control arm. However, there was a high proportion

**Table 1. Participants' socio-demographic characteristics.**

| Variable | Overall (n = 330) | Waiting list control (n = 172) | Intervention arm (n = 158) | Difference statistics and p-value |
|---|---|---|---|---|
| Age (median [IQR]) | 24 (26.0-22.0) | 24.4 (26.3-22.0) | 24.0 (26.0-22.0) | W = 13926, p = 0.695 |
| **Town** | | | | |
| Bungoma | 168 (50.9%) | 88 (51.2%) | 80 (50.6%) | $\chi^2$ = 0.009, p = 0.923 |
| Kitale | 162 (49.1%) | 84 (48.8%) | 78 (49.4%) | |
| **Gender** | | | | |
| Female | 154 (46.7%) | 92 (53.5%) | 62 (39.2%) | **$\chi^2$ = 6.717, p = 0.010** |
| Males | 176 (53.3%) | 80 (46.5%) | 96 (60.8%) | |
| **Education** | | | | |
| Primary | 16 (4.8%) | 8 (4.7%) | 8 (5.1%) | **$\chi^2$ = 6.207, p = 0.045** |
| Secondary | 157 (47.6%) | 71(41.3%) | 86 (54.4%) | |
| Tertiary | 157 (47.6%) | 93 (54.1%) | 64 (40.5%) | |
| **Marital status** | | | | |
| Single | 233 (70.6%) | 124 (72.1%) | 109 (69.0%) | $\chi^2$ = 0.497, p = 0.780 |
| Married | 84 (25.5%) | 41 (23.8%) | 43 (27.2%) | |
| Divorced/separated | 13 (3.9%) | 7 (4.1%) | 6 (3.8%) | |
| **Employment** | | | | |
| Employed/self-employed | 166 (50.3%) | 76 (44.2%) | 90 (57.0%) | **$\chi^2$ = 9.300, P = 0.010** |
| Unemployed | 99 (30.0%) | 64 (37.2%) | 35 (22.2%) | |
| Students | 65 (19.7%) | 32 (18.6%) | 33 (20.9%) | |

IQR = Interquartile Range.

**Table 2. Study outcomes at pre-intervention assessment.**

| Outcome | Overall (n = 330) | Waiting list control (n = 172) | Intervention arm (n = 158) | Difference statistics and p-value |
|---|---|---|---|---|
| **Depression level** | | | | |
| None | – | – | – | |
| Mild | 209 (63.3) | 114 (66.3) | 95 (60.1) | χ2 = 1.342, p = 0.247 |
| Moderate | 121 (36.7) | 58 (33.7) | 63 (39.9) | |
| Severe | – | – | – | |
| **Anxiety** | | | | |
| None | 41 (12.4) | 24 (14.0) | 17 (10.8) | χ2 = 2.983, p = 0.394 |
| Mild | 162 (49.1) | 89 (51.7) | 73 (46.2) | |
| Moderate | 86 (26.1) | 41 (23.8) | 45 (28.5) | |
| Severe | 41 (12.4) | 18 (10.5) | 23 (14.5) | |
| **Stress level** | | | | |
| Low | 9 (2.7) | 7 (4.1) | 2 (1.3) | χ2 = 18.958, p < 0.001 |
| Moderate | 225 (68.2) | 99 (57.5) | 126 (79.7) | |
| High | 96 (29.1) | 66 (38.4) | 30 (19.0) | |
| **Resilience** | | | | |
| Low | 184 (55.8) | 99 (57.6) | 85 (53.8) | χ2 = 2.504, p = 0.286 |
| Normal | 144 (43.6) | 71 (41.3) | 73 (46.2) | |
| High | 2 (0.6) | 2 (1.2) | 0 (0) | |

The median pre-intervention WHODAS score for the control group (27, IQR 30–21) was, higher compared to that of the intervention group (23, IQR = 28–20), p = 0.014. (Fig 2).

of significant improvement in resilience among participants at the control arm (12.0%) compared to participants at the intervention arm (3.5%). Besides, there was high proportion of participants whose resilience worsened in the intervention group (40.3%) compared to the control group (17.6%) (Table 3).

On univariate analysis, only functioning impairment did not significantly improve post intervention. Over time at post-intervention, the intervention group participants had 95%, 82%, and 72% lower odds of having higher depression (OR=0.05, 95% CI 0.02-0.11) anxiety (OR= 0.18, 95% CI 0.09-0.34) and stress (OR=0.28, 95% CI 0.11-0.69) levels, respectively and the association was significant (p<0.05). Participants under the intervention group had 85% lower odds of having higher self-identified problem impact scores (OR=0.15, 95% CI 0.08-0.28, P<0.001) and 4.2 times higher odds of having higher resilience scores (OR=4.22, 95% CI 1.95-9.13, P<0.001) compared to the control group participants (Table 4).

## Adjusted intervention effect

After adjusting for town of residence, gender, education, and employment status and random cluster effects, the intervention group participants had 95%, 82% and 71% lower odds of higher level of depression (AOR=0.05, 95% CI 0.02 -0.10, p<0.001), anxiety (AOR=0.18, 95% CI 0.10–0.35, p<0.001) and stress (AOR=0.29, 95% CI 0.12 – 0.71, p=0.007) compared to those in the control group, respectively. They also had 86% lower odds of higher self-identified problem impact scores (AOR=0.14, 95% CI = 0.08 -0.27, p<0.001) and 4.3 times higher odds of higher resilience levels (AOR=4.33, 95%CI 2.00 –9.36, p<0.001) compared to the youths in the control group (Table 5).

On multiple imputation of missing data, the youth in the intervention group had 92%, 78%, and 76% lower odds of having high level of depression (AOR=0.08, 95%CI 0.04-0.17, p<0.001), anxiety (AOR=0.22, 95%CI 0.11-0.42, p<0.001) and stress (AOR=0.24, 95% CI 0.09-0.64, p=0.005), respectively compared to those in the control group. At post-intervention, youth in the intervention group had 60% lower odds of high level of functioning impairment (AOR=0.40, 95% CI 0.19-0.86, p=0.019) and 84% lower odds of high self-identified problem impact scores (AOR=0.16 (0.09-0.31,

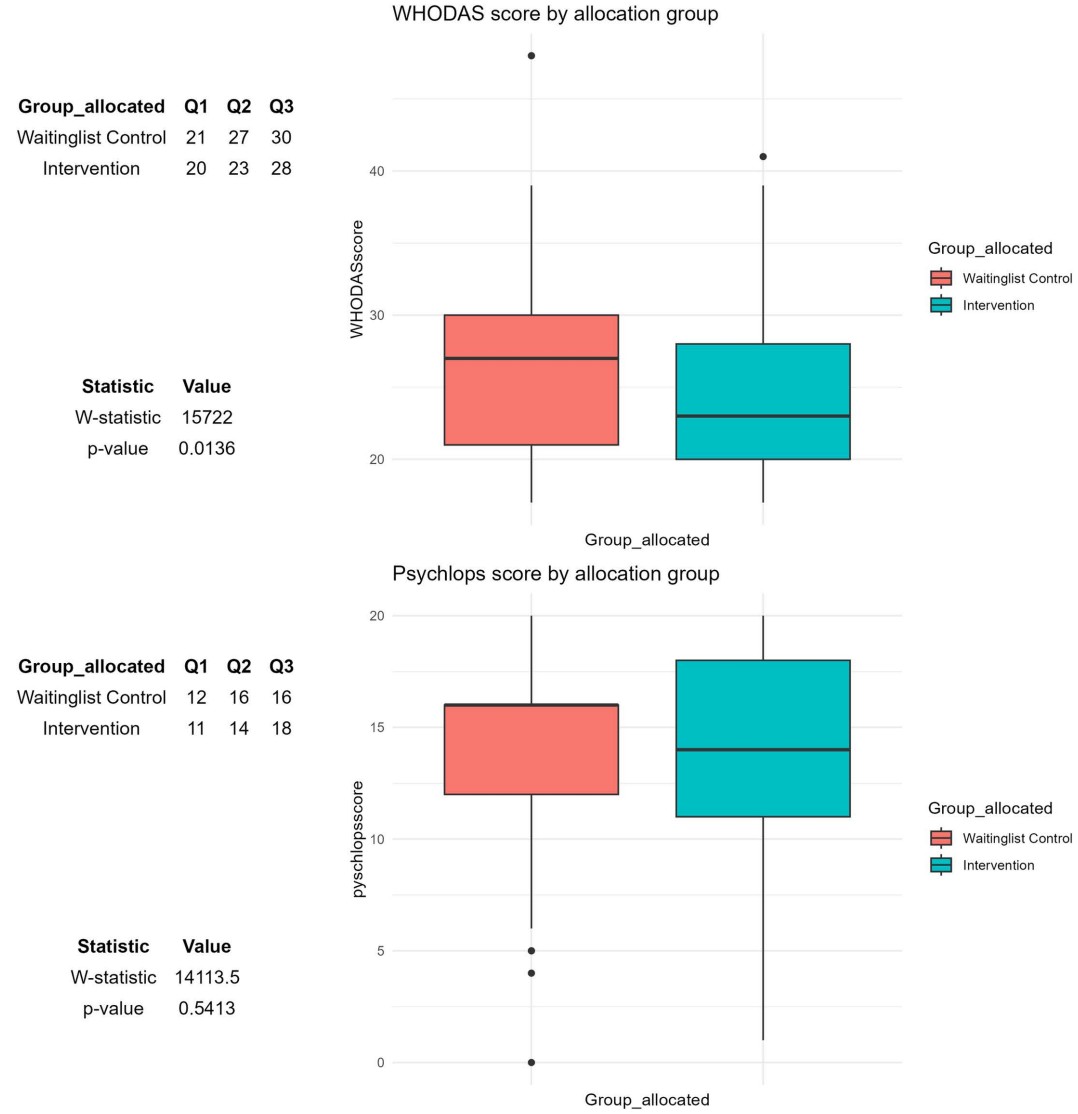

**Fig 2. Comparison of WHODAS and PSYCHLOPS scores between intervention and waiting-list control groups.** Note: Q1 = Lower Quartile, Q2 = Median, Q3 = Upper quartile.

p < 0.001) compared to those in the control group. Participants who received the intervention had 4.35 times higher odds of having high levels of resilience compared to those in the control group (AOR = 4.35, 95%CI 2.03- 9.32, < 0.001). Hence, the results from the multiple imputation data were nearly similar to those from the original data analysis except for functioning impairment where the association become statistically significant. (Table 6).

The results from the ordinal model using the Bayesian regression were not significantly different from those from the clmm model (Table 7).

Models 4, adjacent-category model without category-specific effects (Models in S6 File) results for each outcome were not significantly different from models 1 (Bayesian cumulative ordinal model with Proportional odds assumption) (Models in S1 File) results for the respective outcome. There were slight changes in AORs for Models 3 (Unequal variances

**Table 3. Absolute significant changes in outcome after the intervention period.**

| Significant changes on outcomes | Total (n = 269) | Control (n = 125) | Intervention (n = 144) | Difference statistics |
|---|---|---|---|---|
| **Depression (n = 270)** | | | | |
| No change | 117 (43.3%) | 77 (61.1%) | 40 (27.8%) | χ2 = 52.333, P < 0.001 |
| Worsening | 32 (11.9%) | 22 (17.5%) | 10 (6.9%) | |
| Improvement | 121 (44.8%) | 27 (21.4%) | 94 (65.3%) | |
| **Anxiety (n = 269)** | | | | |
| No change | 111 (41.3%) | 61 (48.8%) | 50 (34.7%) | χ2 = 18.722, p < 0.001 |
| Worsening | 46 (17.1%) | 29 (23.2%) | 17 (11.8%) | |
| Improvement | 112 (41.6%) | 35 (28.0%) | 77 (53.5%) | |
| **Stress (n = 269)** | | | | |
| No change | 179 (66.5%) | 84 (67.2%) | 95 (66.0%) | χ2 = 6.670, p = 0.036 |
| Worsening | 11 (4.1%) | 9 (7.2%) | 2 (1.4%) | |
| Improvement | 79 (29.4%) | 32 (25.6%) | 47 (32.6%) | |
| **Resilience (n = 269)** | | | | |
| No change | 169 (62.8%) | 88 (70.4%) | 81 (56.2%) | χ2 = 20.249, p < 0.001 |
| Worsening | 80 (29.7%) | 22 (17.6%) | 58 (40.3%) | |
| Improvement | 20 (7.4%) | 15 (12.0%) | 5 (3.5%) | |

**Table 4. Unadjusted ordinal cumulative link mixed models (CLMM) results for the association between study outcomes and intervention, accounting for clustering.**

| | Depression | | Anxiety | | Stress | |
|---|---|---|---|---|---|---|
| Variable | Crude OR (95% CI) | P value | Crude OR (95% CI) | P value | Crude OR (95% CI) | P value |
| Time: post | 0.97 (0.60-1.58) | 0.911 | 0.99 (0.63-1.55) | 0.952 | 0.30 (0.17-0.53) | 0.000 |
| Group allocated: Intervention | 1.04 (0.44-2.42) | 0.935 | 0.82 (0.24-2.74) | 0.746 | 0.63 (0.18-2.17) | 0.467 |
| Time: Post*Group allocated: Intervention | 0.05 (0.02-0.11) | 0.000 | 0.18 (0.09-0.34) | 0.000 | 0.28 (0.11-0.69) | 0.006 |
| | **Functioning** | | **Self-identified problem** | | **Resilience** | |
| Variable | Crude OR (95% CI) | P value | Crude OR (95% CI) | P value | Crude OR (95% CI) | P value |
| Time: post | 0.18 (0.10-0.34) | 0.000 | 0.54 (0.35-0.83) | 0.005 | 1.49 (0.86-2.60) | 0.154 |
| Group allocated: Intervention | 0.86 (0.30-2.46) | 0.777 | 0.72 (0.26-2.01) | 0.534 | 1.17 (0.34-4.03) | 0.805 |
| Time: Post*Group allocated: Intervention | 0.54 (0.25-1.14) | 0.104 | 0.15 (0.08-0.28) | 0.000 | 4.22 (1.95-9.13) | 0.000 |

**Reference Group:** Time:pre-, Group allocated: Waiting list control.

model with unequal variance in selected variables) (Models in S3 File), Model 3a (Unequal variances model with unequal variance in all covariates) (Models in S4 File), Model 3b (Unequal variances model with unequal variance in all covariates + Interaction term) (Models in S5 File) which did not lead to changes in outcome effect size direction.

Removing clusters with less than 5 people per cluster did not result in the change of estimate direction or significant of association despite effect size change for stress of 12.1% from 0.29 to 0.33 and impact of self-identified problem of 12.5% from 0.14 to 0.16. This was also the case when all clusters with less than 10 people were removed, with effect size change of more than 10% only observed for anxiety with and effect size estimate change of 12.5% from AOR of 0.18 to AOR of 0.16. All the other outcomes estimate remained stable (Fig 3).

**Table 5. Ordinal cumulative link mixed models (CLMM) results for adjusted association between study outcomes and intervention.**

| Variable | Depression | | Anxiety | | Stress | |
|---|---|---|---|---|---|---|
| | AOR (95% CI) | P value | AOR (95% CI) | P value | AOR (95% CI) | P value |
| Time: Post Intervention | 0.99 (0.61- 1.60) | 0.959 | 0.97 (0.62-1.53) | 0.905 | 0.28 (0.16-0.50) | <0.001 |
| Group allocated: Intervention | 0.98 (0.48-1.98) | 0.950 | 0.78 (0.28-2.18) | 0.640 | 0.61 (0.23-1.62) | 0.321 |
| Gender: Female | 1.20 (0.85-1.70) | 0.305 | 1.30 (0.93-1.81) | 0.127 | 1.78 (1.17-2.72) | 0.008 |
| Education level: Secondary | 1.17 (0.51-2.67) | 0.704 | 1.34 (0.60-2.99) | 0.483 | 0.67 (0.27-1.66) | 0.385 |
| Education level: Tertiary | 1.14 (0.49-2.63) | 0.759 | 1.59 (0.70-3.62) | 0.266 | 0.85 (0.34-2.15) | 0.737 |
| Town: Kitale | 0.47 (0.24-0.93) | 0.030 | 0.23 (0.08-0.66) | 0.006 | 0.23 (0.09-0.59) | 0.002 |
| Employment status: Unemployed | 0.77 (0.51-1.16) | 0.218 | 0.91 (0.62-1.34) | 0.626 | 1.64 (1.00-2.67) | 0.048 |
| Employment status: Students | 0.64 (0.39-1.06) | 0.084 | 0.84 (0.52-1.37) | 0.482 | 0.80 (0.43-1.50) | 0.481 |
| Time: Post*Group allocated: Intervention | 0.05 (0.02-0.10) | <0.001 | 0.18 (0.09-0.34) | <0.001 | 0.29 (0.12-0.71) | 0.007 |
| **Variable** | **Functioning** | | **Self-identified problem** | | **Resilience** | |
| | AOR (95% CI) | P value | AOR (95% CI) | P value | AOR (95% CI) | P value |
| Time: Post Intervention | 0.17 (0.10-0.32) | <0.001 | 0.54 (0.35-0.84) | 0.006 | 1.48 (0.85-2.56) | 0.163 |
| Group allocated: Intervention | 0.95 (0.46-1.93) | 0.877 | 0.71 (0.31-1.62) | 0.419 | 1.29 (0.55-3.04) | 0.556 |
| Gender: Female | 1.59 (1.07-2.36) | 0.021 | 1.26 (0.91 -1.74) | 0.173 | 1.21 (0.81-1.81) | 0.353 |
| Education level: Secondary | 0.91 (0.37-2.25) | 0.834 | 0.87 (0.42-1.78) | 0.697 | 1.20 (0.47-3.02) | 0.706 |
| Education level: Tertiary | 0.87 (0.35-2.18) | 0.765 | 0.92 (0.44-1.92) | 0.824 | 1.22 (0.47-3.14) | 0.680 |
| Town: Kitale | 6.21 (3.15-12.26) | <0.001 | 0.29 (0.13-0.65) | 0.003 | 6.67 (2.87-15.50) | <0.001 |
| Employment status: Unemployed | 0.89 (0.56-1.41) | 0.609 | 1.59 (1.09-2.34) | 0.017 | 1.19 (0.75-1.90) | 0.456 |
| Employment status: Students | 0.86 (0.49-1.52 | 0.600 | 0.89 (0.56-1.41) | 0.616 | 0.88 (0.50-1.57) | 0.675 |
| Time: Post*Group allocated: Intervention | 0.55 (0.26-1.16) | 0.115 | 0.14 (0.08-0.27) | <0.001 | 4.33 (2.00-9.36) | <0.001 |

**Reference group** Time:pre, Group allocated: Waitinglist control, Gender: Male, Education: Primary, Town: Bungoma, Employment: Employed/self-employed.

## Discussion

Based on our context, barbers are primarily young people with whom the youth can relate. Besides, barbers are trusted by their customers. Many young people in Western Kenya spend some of their free time in barbershops seeking haircuts or discussing topics like football, politics, and current affairs. With the rapport already created, stigma barrier against mental health is minimised, as the barbers can freely discuss about mental health with the young people.

The study findings show that DWM delivered by barbers, as lay helpers, reduces psychological distress as defined in our study by levels of depression, anxiety and self-perceived stress among literate youth. It also resulted in a significant improvement of resilience, but had little to no effect on their functioning. These findings are consistent with that of Ataturk et al. (2020) among Turkish nationals and Syrian refugee adults with psychological distress, where DWM resulted in decreased mental health distress [18]. It is also consistent with the indicative benefits of DWM in reducing stress, anxiety levels and personally-identified problems among internally displaced persons, returnees and host community teens and adults in Zummar, Iraq [13], even though that study did not have a control group.

The intervention showed mixed significance of effect on functioning impairment among the participants. Relatable to our study findings, a previous study utilising the broader SH+ intervention did not find significant improvement in functioning among the participants [14]. With functioning impairment being a result of multifaceted causes, including physical health, Acceptance and Commitment Therapy (ACT) based community mental health interventions, like DWM, might not directly address all components of functioning impairment, since it may be linked to external multifaceted factors, such as unemployment resulting from non-psychological causes.

**Table 6. Ordinal cumulative link mixed models (CLMM) results for adjusted regression between study outcomes and time: intervention group based on multiple imputed data.**

| Variable | Depression AOR (95% CI) | P value | Anxiety AOR (95% CI) | p value | Stress AOR (95% CI) | p value |
|---|---|---|---|---|---|---|
| Time: Post Intervention | 0.90 (0.56-1.44) | 0.647 | 0.96 (0.61-1.53) | 0.873 | 0.47 (0.24-0.89) | 0.022 |
| Group allocated: Intervention | 1.00 (0.51-1.97) | 1.000 | 0.88 (0.34-2.29) | 0.793 | 0.64 (0.24-1.73) | 0.384 |
| Gender: Female | 1.14 (0.81-1.60) | 0.470 | 1.26 (0.91-1.74) | 0.164 | 1.65 (1.09-2.49) | 0.017 |
| Education level: Secondary | 1.10 (0.48-2.52) | 0.817 | 1.29 (0.58-2.86) | 0.526 | 0.79 (0.31-2.01) | 0.626 |
| Education level: Tertiary | 1.09 (0.47-2.51) | 0.843 | 1.51 (0.67-3.39) | 0.315 | 0.98 (0.38-2.49) | 0.963 |
| Town: Kitale | 0.49 (0.26-0.94) | 0.032 | 0.25 (0.09-0.66) | 0.006 | 0.20 (0.07-0.53) | 0.001 |
| Employment status: Unemployed | 0.80 (0.54-1.19) | 0.264 | 0.94 (0.65-1.38) | 0.762 | 1.64 (1.02-2.63) | 0.044 |
| Employment status: Students | 0.71 (0.44-1.16) | 0.170 | 0.91 (0.56-1.48) | 0.700 | 0.92 (0.49-1.71) | 0.788 |
| Time: Post*Group allocated: Intervention | 0.08 (0.04-0.17) | 0.000 | 0.22 (0.11-0.42) | 0.000 | 0.24 (0.09-0.64) | 0.005 |

| Variable | Functioning impairment AOR (95% CI) | p value | Self-identified problem AOR (95% CI) | p value | Resilience AOR (95% CI) | p value |
|---|---|---|---|---|---|---|
| Time: Post Intervention | 0.32 (0.17-0.58) | 0.000 | 0.62 (0.39-0.96) | 0.034 | 1.29 (0.74-2.23) | 0.366 |
| Group allocated: Intervention | 1.01 (0.49-2.08) | 0.981 | 0.79 (0.37-1.68) | 0.537 | 1.24 (0.53-2.92) | 0.625 |
| Gender: Female | 1.49 (1.01-2.20) | 0.046 | 1.19 (0.86-1.63) | 0.289 | 1.21 (0.81-1.80) | 0.344 |
| Education level: Secondary | 1.04 (0.42-2.55) | 0.935 | 0.90 (0.44-1.83) | 0.778 | 1.15 (0.46-2.90) | 0.766 |
| Education level: Tertiary | 0.98 (0.40-2.42) | 0.963 | 0.96 (0.46-1.98) | 0.911 | 1.17 (0.45-3.04) | 0.740 |
| Town: Kitale | 4.43 (2.23-8.82) | 0.000 | 0.29 (0.14-0.61) | 0.001 | 6.94 (2.93-16.43) | 0.000 |
| Employment status: Unemployed | 0.90 (0.57-1.41) | 0.649 | 1.55 (1.06-2.24) | 0.022 | 1.14 (0.72-1.81) | 0.564 |
| Employment status: Students | 0.93 (0.54-1.61) | 0.790 | 0.96 (0.61-1.51) | 0.847 | 0.87 (0.49-1.54) | 0.624 |
| Time: Post*Group allocated: Intervention | 0.40 (0.19-0.86) | 0.019 | 0.16 (0.09-0.31) | 0.000 | 4.35 (2.03-9.32) | 0.000 |

**Reference Group** Time:pre, Group allocated: Waitinglist control, Gender: Male, Education: Primary, Town: Bungoma, Employment: Employed/self-employed.

**Table 7. Exponentiated Estimate from the Cumulative ordinal model with equal variance and proportional odds assumption fitted using Bayesian ordinal multilevel regression (Models 1).**

| Predictor | depression AOR (95% CI) | Anxiety AOR (95% CI) | Stress AOR (95% CI) | Functioning AOR (95% CI) | Self-identified problem AOR (95% CI) | Resilience AOR (95% CI) |
|---|---|---|---|---|---|---|
| Time: Post Intervention | 0.99 (0.61-1.60) | 0.98 (0.62-1.54) | 0.28 (0.16-0.51) | 0.17 (0.09-0.32) | 0.54 (0.35-0.84) | 1.48 (0.84-2.59) |
| Group allocated: Intervention | 0.97 (0.40-2.44) | 0.77 (0.22-2.56) | 0.60 (0.19-1.92) | 0.94 (0.40-2.25) | 0.70 (0.25-1.94) | 1.35 (0.46-4.24) |
| Gender: Female | 1.20 (0.85-1.71) | 1.30 (0.93-1.83) | 1.80 (1.17-2.81) | 1.59 (1.05-2.40) | 1.27 (0.92-1.76) | 1.23 (0.82-1.85) |
| Education level: Secondary | 1.17 (0.50-2.75) | 1.33 (0.58-3.01) | 0.66 (0.27-1.68) | 0.95 (0.38-2.37) | 0.86 (0.41-1.81) | 1.22 (0.49-3.17) |
| Education level: Tertiary | 1.14 (0.48-2.67) | 1.60 (0.69-3.71) | 0.85 (0.33-2.17) | 0.90 (0.35-2.28) | 0.92 (0.43-1.92) | 1.27 (0.48-3.36) |
| Town: Kitale | 0.48 (0.20-1.13) | 0.24 (0.07-0.83) | 0.23 (0.07-0.75) | 6.63 (2.96-15.83) | 0.29 (0.11-0.82) | 7.45 (2.56-26.29) |
| Employment status: Unemployed | 0.77 (0.51-1.16) | 0.91 (0.61-1.33) | 1.64 (0.99-2.70) | 0.88 (0.56-1.39) | 1.59 (1.08-2.33) | 1.19 (0.74-1.92) |
| Employment status: Students | 0.65 (0.39-1.09) | 0.83 (0.51-1.34) | 0.81 (0.43-1.54) | 0.88 (0.50-1.55) | 0.88 (0.56-1.40) | 0.89 (0.50-1.59) |
| Time: Post*Group allocated: Intervention | 0.05 (0.02-0.10) | 0.18 (0.09-0.34) | 0.27 (0.11-0.67) | 0.55 (0.26-1.17) | 0.14 (0.07-0.26) | 4.43 (2.07-9.68) |

**Reference Group** Time:pre, Group allocated: Waitinglist control, Gender: Male, Education: Primary, Town: Bungoma, Employment: Employed/self-employed.

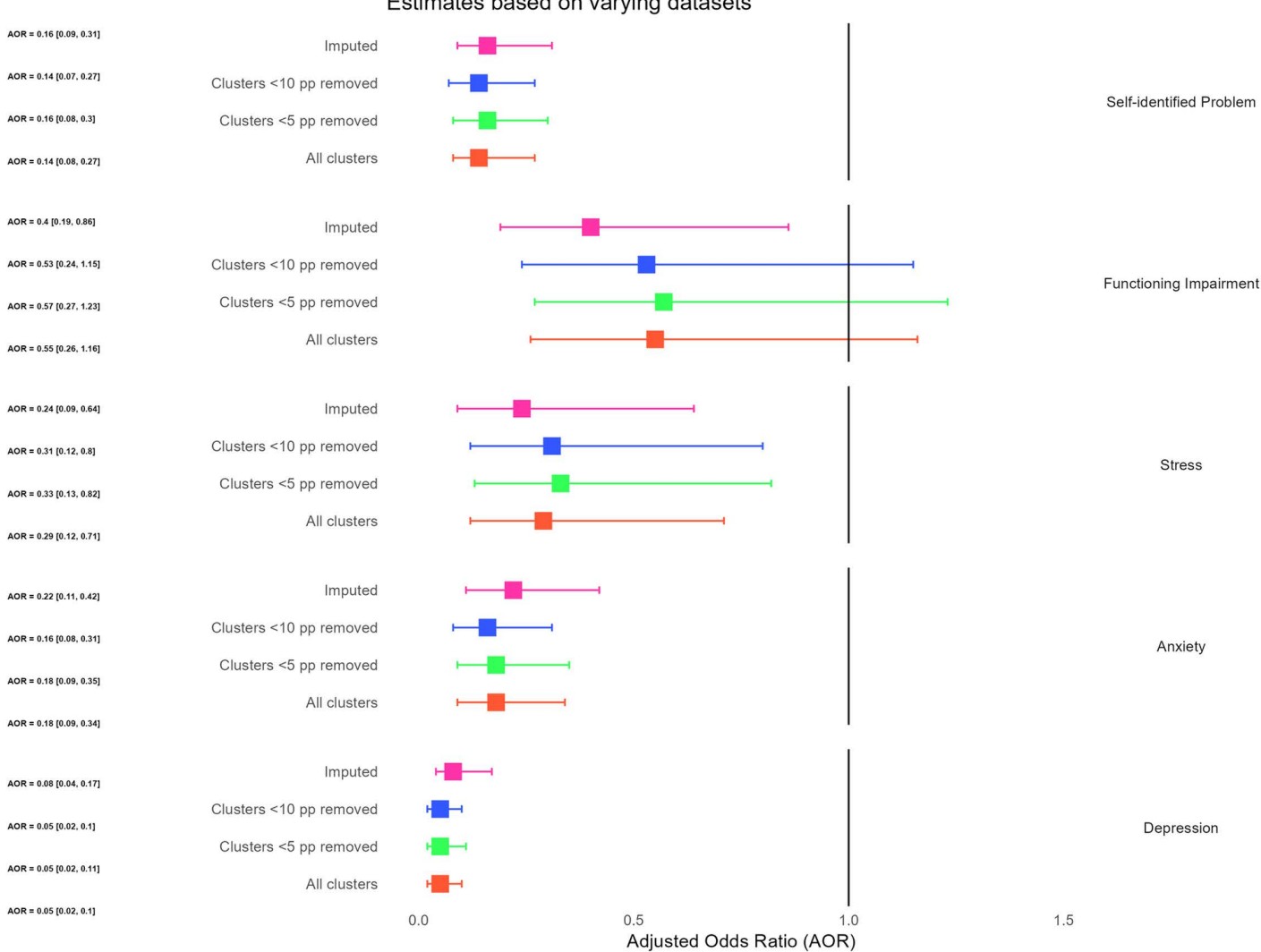

**Fig 3. Comparison of models estimates based on varying minimum participants per cluster and imputed data.** Note Fig 3: Imputed: Represents results from the model based on imputed data. Clusters <10 pp removed: Represents results of the model when clusters with fewer than 10 people each are excluded from analysis. Clusters <5 pp removed represents results of the model when clusters with fewer than 5 people each are excluded from analysis. All clusters: Represent results of the original model based on all clusters and participants in the study.

It is evident that trained lay providers can deliver DWM to help alleviate psychological distress among young people. In a short period of time (five weeks), there was observed an immediate effect of the intervention. While barbers have been widely utilised in the delivery of health education and promotion interventions among African minority groups in the United States [45–47], our study demonstrates the feasibility of utilising barbers in the Kenyan context to deliver community-based mental health intervention with a high reach. Moreover, the collective composition of mental health-themed songs as a raising awareness tool and an entry point to the intervention is an overriding community innovation that the study introduced.

Besides, the recruitment and capacity building of barbers consist an innovative approach to reaching youth in their own comfort environment and establishing trusty relationships with them within a short period of time. This is likely to

have played a role in the youths' participation and observed positive study outcomes. Furthermore, barbers were peer young people, who may also be considered as indirect beneficiaries of the project. Alongside their capacity building on basic mental health skills, they were likely to continue applying them in their daily lives as needed with a sustainable impact.

The intervention's effectiveness observed in our study may partly be attributed to the high literacy level of our study population. Only education level was significantly associated with the study outcomes of anxiety and self-perceived stress in the adjusted models. The level of reduction in anxiety and stress symptoms was positively associated with higher levels of education. ACT-based interventions, such as Problem Management Plus (PM+), are highly effective among individuals with higher levels of education [48,49]. Further, contextual adaptation of the intervention, including translation of intervention materials into local languages and dialects, might help increase its applicability and use among illiterate youths.

Despite the multiple access barriers to mental health services in Western Kenya, including stigma, limited investment in mental health services and few mental health specialists [9], our study findings showed that it is possible to address these challenges through the provision of low-cost, accessible and youth-led psychological interventions. Our study demonstrates that DWM can be utilised effectively as a standalone package, independent to the broader SH+ package, in addressing community mental health needs among young people. It further contributes in youth-friendly mental health promotion and prevention of severe mental disorders. Further, it is among the first studies to our knowledge, to utilise lay-supported DWM intervention for youth, showing that this intervention is applicable in low-resource contexts, beyond humanitarian settings.

Our study raises some issues for future research. Multisite studies with representative samples are needed to determine the external validity and replicability of the DWM intervention for young people. This can later inform its wider community-based dissemination and utilisation for the youth in Kenya. Besides, an economic evaluation of the intervention is needed to provide an economic case for its wide adoption. The study was limited to literate youths, which may pose important concerns for generalisability to illiterate youths.

As part of future work, we will seek to understand barbers' experiences delivering the intervention and project staff reflections on the intervention delivery process.

## Limitations

We note several limitations of our study. First, it was powered to detect medium effect sizes rather than small ones. This limits the extrapolation of the study findings. Second, we only did an assessment at the end of the intervention delivery; hence, it is not possible to determine if the observed effect is sustained in the long term.

## Conclusion

Our study indicates that barbershops are likely viable community places to promote mental health awareness and prevent mental disorders through the provision of low-intensity mental health interventions. Barber-facilitated DWM in Western Kenya is a culturally acceptable, youth-friendly, feasible and potentially safe low-intensity intervention that promotes mental health and can prevent severe mental health disorders. It is an effective first-line community-based intervention for young people experiencing psychological distress in Western Kenya and similar settings. Its implementation requires adequate capacity building and supervision by trained mental health specialists to address any potential adverse events and an integrated stepped-care framework where those with severe mental health disorders are referred to appropriate clinical interventions. Our study results underscore the potential of leveraging culturally embedded, community-based platforms to deliver scalable, youth-friendly mental health interventions, thereby contributing to the reduction of the mental health treatment gap in low-resource settings like Western Kenya.

## Supporting information

**S1 Text. Tool for assessment of thoughts of suicide and impairment possibly due to severe mental, neurological or substance use disorder.**
(DOCX)

**S2 Text. Intervention procedure.** It provides detailed information on the study procedure.
(DOCX)

**S3 Text. List of outcomes measures and their scoring.** Provides information on all outcome measures used in the study, their validation and scoring.
(DOCX)

**S1 Table. Sample size calculation table.** It presents the results from the sample size calculation indicating the required clusters and participants per cluster.
(DOCX)

**S2 Table. LOOIC from the Bayesian multilevel ordinal regression models.** This table presents Leave-One-Out Information Criterion (LOOIC) values for each Bayesian multilevel ordinal regression model. These values were used to evaluate best fit model for our data.
(DOCX)

**S1 File. Zipped folder with Excel files: Anxiety_model1_results.xlsx; Depression_model1_results.xlsx; Problem_model1_results.xlsx; Resilience_model1_results.xlsx; Stress_model1_results.xlsx; WHODAS_model1_results.xlsx.** Collectively referred to as Models 1. They contain results of Bayesian cumulative ordinal model with proportional odds assumption for each study outcome.
(ZIP)

**S2 File. Zipped folder with Excel files: Anxiety_model2_results.xlsx; Depression_model2_results.xlsx; Problem_model2_results.xlsx; Resilience_model2_results.xlsx; Stress_model2_results.xlsx; WHODAS_model2_results.xlsx.** Collectively referred to as Models 2. They contain results of the category-specific effects model using adjacent-category method for each of the study outcome.
(ZIP)

**S3 File. Zipped folder with Excel files: Anxiety_model3_results.xlsx; Depression_model3_results.xlsx; Problem_model3_results.xlsx; Resilience_model3_results.xlsx; Stress_model3_results.xlsx; WHODAS_model3_results.xlsx.** Collectively referred to as Models 3. They contain results of the unequal variances model with unequal variance in selected covariates for each of the study outcome.
(ZIP)

**S4 File. Zipped folder with Excel files: Anxiety_model3a_results.xlsx; Depression_model3a_results.xlsx; Problem_model3a_results.xlsx; Resilience_model3a_results.xlsx; Stress_model3a_results.xlsx; WHODAS_model3a_results.xlsx.** Collectively referred to as Models 3a. Contains results of the unequal variances model with unequal variance in all covariates for each of the study outcome.
(ZIP)

**S5 File. Zipped folder with Excel files: Anxiety_model3b_results.xlsx; Depression_model3b_results.xlsx; Problem_model3b_results.xlsx; Resilience_model3b_results.xlsx; Stress_model3b_results.xlsx; WHODAS_model3b_results.xlsx.** Collectively referred to as Models 3b. Contains results of the unequal variances model with unequal variance in all covariates +interaction term.
(ZIP)

**S6 File. Zipped folder with Excel files: Anxiety_model4_results.xlsx; Depression_model4_results.xlsx; Problem_model4_results.xlsx; Resilience_model4_results.xlsx; Stress_model4_results.xlsx; WHODAS_model4_results.xlsx.** Collectively referred to as Models 4. Contains results of the adjacent-category model without category-specific effects.
(ZIP)

**S7 File. Study data and R codebooks.** It contains; Analysis_codebook.Rmd, which is the r analysis codebook; Imputation_codebook.Rmd, with the imputation process and analysis r codes; and Wematter_data.rds, which is an R format dataset used in the analysis.
(ZIP)

**S1 Checklist. CONSORT 2010 checklist for reporting of a cluster randomised trial.**
(DOCX)

**S1 Protocol. Study protocol.**
(PDF)

## Acknowledgments

We thank the barbers who participated in delivering the intervention and the team of Research Assistants involved in the project monitoring and evaluation.

## Author contributions

**Conceptualization:** Protus Musotsi Yabunga, Phiona Naserian Koyiet, Ken Simiyu, Mary Nangukhula.

**Data curation:** Protus Musotsi Yabunga, Harriet Musimbi, Ken Simiyu, Peter Koskei.

**Formal analysis:** Protus Musotsi Yabunga, Peter Koskei.

**Funding acquisition:** Protus Musotsi Yabunga, Mary Nangukhula.

**Investigation:** Protus Musotsi Yabunga, Phiona Naserian Koyiet, Harriet Musimbi, Ken Simiyu, Elijah Chemorei, Mary Nangukhula.

**Methodology:** Protus Musotsi Yabunga, Phiona Naserian Koyiet, Ken Simiyu, Peter Koskei.

**Project administration:** Protus Musotsi Yabunga, Harriet Musimbi, Ken Simiyu, Elijah Chemorei, Mary Nangukhula.

**Resources:** Protus Musotsi Yabunga, Phiona Naserian Koyiet, Ken Simiyu, Mary Nangukhula.

**Software:** Protus Musotsi Yabunga.

**Supervision:** Protus Musotsi Yabunga, Phiona Naserian Koyiet, Harriet Musimbi, Ken Simiyu, Elijah Chemorei, Mary Nangukhula.

**Validation:** Protus Musotsi Yabunga, Phiona Naserian Koyiet, Ken Simiyu, Peter Koskei, Stella Evangelidou.

**Visualization:** Protus Musotsi Yabunga, Ken Simiyu, Stella Evangelidou.

**Writing – original draft:** Protus Musotsi Yabunga.

**Writing – review & editing:** Protus Musotsi Yabunga, Phiona Naserian Koyiet, Harriet Musimbi, Ken Simiyu, Elijah Chemorei, Peter Koskei, Stella Evangelidou, Mary Nangukhula.

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
