## [Decision Letter · Decision Letter 0]

24 Sep 2024

PGPH-D-24-01544

Effectiveness of Barber-Facilitated "Doing What Matters in Times of Stress" Intervention Among Urban Literate Youths in Western Kenya: A cluster randomised trial

Dear Dr. Yabunga,

Thank you for submitting your manuscript to PLOS Global Public Health. After careful consideration, we feel that it has merit but does not fully meet PLOS Global Public Health’s publication criteria as it currently stands. Therefore, we invite you to submit a revised version of the manuscript that addresses the points raised during the review process.

Please note that we have only been able to secure a single reviewer to assess your manuscript. We are issuing a decision on your manuscript at this point to prevent further delays in the evaluation of your manuscript. Please be aware that the editor who handles your revised manuscript might find it necessary to invite additional reviewers to assess this work once the revised manuscript is submitted. However, we will aim to proceed on the basis of this single review if possible. 

We look forward to receiving your revised manuscript.

Kind regards,

Johanna Pruller, Ph.D.

Staff Editor

Journal Requirements:

1. Your current Financial Disclosure does not have Grant Recipient. However, your funding information on the submission form indicates Grant Recipient “Protus Musotsi Yabunga". Please indicate by return email the full and correct funding information for your study and confirm the order in which funding contributions should appear. Please be sure to indicate whether the funders played any role in the study design, data collection and analysis, decision to publish, or preparation of the manuscript.

2. We note that your Data Availability Statement is currently as follows: "The data supporting the findings of this study will be made available open source. We are working on putting it on an open data repository."

3. Please provide separate figure files in .tif or .eps format.

Additional Editor Comments (if provided):

Reviewers' comments:

Reviewer's Responses to Questions

**Comments to the Author**

1. Does this manuscript meet PLOS Global Public Health’s publication criteria ? Is the manuscript technically sound, and do the data support the conclusions? The manuscript must describe methodologically and ethically rigorous research with conclusions that are appropriately drawn based on the data presented.

Reviewer #1: Partly

2. Has the statistical analysis been performed appropriately and rigorously?

Reviewer #1: No

3. Have the authors made all data underlying the findings in their manuscript fully available (please refer to the Data Availability Statement at the start of the manuscript PDF file)?

Reviewer #1: No

4. Is the manuscript presented in an intelligible fashion and written in standard English?

Reviewer #1: Yes

5. Review Comments to the Author

Reviewer #1: The manuscript analyses original data through a rather widely-used approach. Though the research questions are well posed, I cast some doubts about the employed methods. More details and sensitivity analyses are require in general. Detailed comments follow.

1. The data and the code used to perform the analysis are not attached to the manuscript. Please, add both as supplementary material to ensure the reproducibility of the results.

2. Up to what I understand, the outcomes are measured on a discrete scale. If so, it is rather unclear to me why linear models are considered, as they are thought for continuous variables. I suggest to look at generalized linear (mixed) models instead; these would better reflect the proposed categorization described from line 290, where essentially categorical ordinal outcomes are defined.

3. Bearing in mind my previous comment, no matter which modelling or testing approaches are considered, the assumptions on which the employed methods are based should be carefully checked. For example, if a t-test is applied, please provide evidence that outliers are not present, homoscedasticity is met, gaussianity is a plausible assumption, etc; similarly, in a regression context the Gauss-Markov assumptions must be checked. In general, more details are required. Is the random effects distribution Gaussian? If so, how many quadrature points are considered? What's the estimate of the random effects variance? Is this assumption plausible? Why random slopes are not considered? Are you sure that all the fitted values are between the minimum and the maximum plausible values? Please, justify all the modelling choices.

4. Complete cases analysis is known to produce biased estimates. I am a bit puzzled on the your statement that missingness is completely non-at-random, how do you test it? I feel that methods such as pattern mixtures or selection models should be further considered and sensitivity analyses performed. This book could be a good reference https://www.routledge.com/Handbook-of-Missing-Data-Methodology/Molenberghs-Fitzmaurice-Kenward-Tsiatis-Verbeke/p/book/9780367739294?srsltid=AfmBOormTtlUOXkGjISUZcE9WCOAlcFs60X0PSaQktOihmf_Mn_qWYxw, but please be aware that the literature on this matter is massive.

5. I see that multiple imputation is considered instead. More details are required on this as well. How datasets were considered? Did you check for the appropriateness of the imputations? Please, take https://www.jstatsoft.org/article/view/v045i03 as reference.

6. Please, provide the estimated coefficients for the confounders as well. Did you estimate any differences across cities?

6. PLOS authors have the option to publish the peer review history of their article (what does this mean? ). If published, this will include your full peer review and any attached files.

**Do you want your identity to be public for this peer review?** For information about this choice, including consent withdrawal, please see our Privacy Policy .

Reviewer #1: No

---

## [Decision Letter · Decision Letter 1]

26 Mar 2025

PGPH-D-24-01544R1

Effectiveness of Barber-Facilitated "Doing What Matters in Times of Stress" Intervention Among Urban Literate Youths in Western Kenya: A cluster randomised trial

Dear Dr. Yabunga,

Thank you for submitting your manuscript to PLOS Global Public Health. After careful consideration, we feel that it has merit but does not fully meet PLOS Global Public Health’s publication criteria as it currently stands. Therefore, we invite you to submit a revised version of the manuscript that addresses the points raised during the review process.

We look forward to receiving your revised manuscript.

Kind regards,

Joel Msafiri Francis, MD, MS, PhD

Academic Editor

Journal Requirements:

Additional Editor Comments (if provided):

Reviewers' comments:

Reviewer's Responses to Questions

**Comments to the Author**

1. If the authors have adequately addressed your comments raised in a previous round of review and you feel that this manuscript is now acceptable for publication, you may indicate that here to bypass the “Comments to the Author” section, enter your conflict of interest statement in the “Confidential to Editor” section, and submit your "Accept" recommendation.

Reviewer #1: All comments have been addressed

Reviewer #2: All comments have been addressed

2. Does this manuscript meet PLOS Global Public Health’s publication criteria ? Is the manuscript technically sound, and do the data support the conclusions? The manuscript must describe methodologically and ethically rigorous research with conclusions that are appropriately drawn based on the data presented.

Reviewer #1: (No Response)

Reviewer #2: Yes

3. Has the statistical analysis been performed appropriately and rigorously?

Reviewer #1: (No Response)

Reviewer #2: Yes

4. Have the authors made all data underlying the findings in their manuscript fully available (please refer to the Data Availability Statement at the start of the manuscript PDF file)?

Reviewer #1: (No Response)

Reviewer #2: Yes

5. Is the manuscript presented in an intelligible fashion and written in standard English?

Reviewer #1: (No Response)

Reviewer #2: Yes

6. Review Comments to the Author

Reviewer #1: (No Response)

Reviewer #2: 1) Overall Comments:

The use of barbers as facilitators for the Do What Matters intervention represents an innovative and culturally grounded approach to expanding access to mental health support, particularly in contexts where mental health providers are limited. I would like to reiterate the public health significance of this research and its potential application in similar geographic and cultural settings.

General Response to Previous Review:

In reference to the concerns raised during the first review, the author has made commendable efforts in addressing the key issues and has ensured that data is available as recommended. Below are the specific comments:

2) Discussion & Conclusion:

The author concludes that “barbershops are potentially safe and feasible community places to promote mental health awareness and prevent mental disorders...” While DWM intervention appears to be low-intensity and potentially safe, concluding definitively on its safety based solely on the findings from this study warrants caution.

There is literature, including anecdotal and research-based reports (see, for example: Chen X, Stanton B, Li X, Fang X, Lin D. Substance use among rural-to-urban migrants in China: a moderation effect model analysis. Subst Use Misuse. 2008;43(1):105-24. doi: 10.1080/10826080701209077. PMID: 18189208 and Barlow, A., & Hawdon, J. (2015). Sex, Drugs, and Deception: Deviance in the Hair Salon Industry. Deviant Behavior, 37(1), 66–80. https://doi.org/10.1080/01639625.2014.983009), suggesting that barbershops and hair salons may, in some instances, also serve as hotspots sites for activities such as substance use and sex work. Given this context, a more nuanced discussion around the safety of barbershops as intervention spaces is warranted.

To strengthen the conclusion, it would be helpful for the author to indicate whether any aspects of safety—including the potential for such adverse contextual factors—were observed or noted in the study or referenced from the literature. Furthermore, ascertain whether there were any serious adverse events (SAE) reported in the course of the study. Insights in this regard would enhance the credibility of the safety claims, particularly considering that some follow-up visits occurred at locations chosen by the participants in agreement with the barber.

If no data on adverse events is available, this should be acknowledged as a limitation, given its implications for future scale-up. Additionally, any safeguards integrated into the DWM barber training or study monitoring process should be highlighted to reassure readers and stakeholders of the ethical and safety frameworks underpinning the intervention.

3) Statistical Analysis:

The author has addressed several concerns from the initial review. However, with respect to the question about the Missing at Random (MAR) assumption, the opening statement in the response—“We have used intention-to-treat analysis instead”—is misapplied. Intention-to-treat (ITT) is a study design principle that ensures participants are analyzed in the groups to which they were randomized; it is not a statistical method that addresses the missing data mechanism, nor does it justify assumptions regarding MAR.

If the assumption is that data were MAR due to their association with observed covariates, the authors should specify which covariates were predictive of missingness and how this informed the MAR assumption. Clarifying this point would strengthen the robustness of the statistical handling of missing data and provide greater transparency in the analysis.

While the manuscript presents the findings in a reasonably understandable manner, the author should consider expanding the explanation and evidence supporting their analytical choices in the revised version.

4) Editorial Comments:

Lines 237–241 should be moved to follow the paragraph describing the “waiting list arm procedure” (currently ending at line 248). Logically, it would be clearer to describe the control group procedures before discussing data collection and follow-up related to that group.

7. PLOS authors have the option to publish the peer review history of their article (what does this mean? ). If published, this will include your full peer review and any attached files.

**Do you want your identity to be public for this peer review?** For information about this choice, including consent withdrawal, please see our Privacy Policy .

Reviewer #1: No

Reviewer #2: No

---

## [Decision Letter · Decision Letter 2]

9 May 2025

Effectiveness of Barber-Facilitated "Doing What Matters in Times of Stress" Intervention Among Urban Literate Youths in Western Kenya: A cluster randomised trial

PGPH-D-24-01544R2

Dear Mr Yabunga,

We are pleased to inform you that your manuscript 'Effectiveness of Barber-Facilitated "Doing What Matters in Times of Stress" Intervention Among Urban Literate Youths in Western Kenya: A cluster randomised trial' has been provisionally accepted for publication in PLOS Global Public Health.

Best regards,

Joel Msafiri Francis, MD, MS, PhD

Academic Editor

Reviewer Comments (if any, and for reference):

Reviewer's Responses to Questions

**Comments to the Author**

1. If the authors have adequately addressed your comments raised in a previous round of review and you feel that this manuscript is now acceptable for publication, you may indicate that here to bypass the “Comments to the Author” section, enter your conflict of interest statement in the “Confidential to Editor” section, and submit your "Accept" recommendation.

Reviewer #2: All comments have been addressed

2. Does this manuscript meet PLOS Global Public Health’s publication criteria ? Is the manuscript technically sound, and do the data support the conclusions? The manuscript must describe methodologically and ethically rigorous research with conclusions that are appropriately drawn based on the data presented.

Reviewer #2: Yes

3. Has the statistical analysis been performed appropriately and rigorously?

Reviewer #2: Yes

4. Have the authors made all data underlying the findings in their manuscript fully available (please refer to the Data Availability Statement at the start of the manuscript PDF file)?

Reviewer #2: Yes

5. Is the manuscript presented in an intelligible fashion and written in standard English?

Reviewer #2: Yes

6. Review Comments to the Author

Reviewer #2: In the resubmitted version, all the concerns I raised have been adequately addressed.

1) Editorial comments: Contents for the intervention and the control groups have been logically re-organized.

2) Statistical analysis: Information on tests for missingness assumptions and details of the analysis conducted to test for MAR assumptions provided.

3) Discussion & Conclusion: Wording used on safety and feasibility conclusions revised accordingly. Furthermore, the author has included information regarding adverse events monitoring which they found none.

7. PLOS authors have the option to publish the peer review history of their article (what does this mean? ). If published, this will include your full peer review and any attached files.

**Do you want your identity to be public for this peer review?** For information about this choice, including consent withdrawal, please see our Privacy Policy .

Reviewer #2: No
